# Health and Safety Practices as Drivers of Business Performance in Informal Street Food Economies: An Integrative Review of Global and South African Evidence

**DOI:** 10.3390/ijerph22081239

**Published:** 2025-08-08

**Authors:** Maasago Mercy Sepadi, Tim Hutton

**Affiliations:** Tshwane School for Business and Society, Tshwane University of Technology, Ditsela Place, 1204 Park Street, Hatfield, Pretoria 0028, South Africa; hut@mweb.co.za

**Keywords:** street food vendors, Health Belief Model, Balanced Scorecard, business performance, South Africa, LMICs, integrative review

## Abstract

Background: Street food vending provides vital employment and nutrition in low- and middle-income countries (LMICs), but poor health and safety compliance pose significant public health and business risks. Despite growing policy recognition, the link between hygiene practices and vendor performance remains underexplored. Objective: This integrative review examines the influence of health and safety practices on the business performance of informal street food vendors, with a particular focus on both global and South African contexts. Methods: A total of 76 studies published between 2015 and 2025 were retrieved between June 2024 and May 2025 and analyzed using an integrative review methodology. Sources were identified through five major academic databases and grey literature repositories. Thematic synthesis followed PRISMA logic and was guided by the Health Belief Model (HBM) and Balanced Scorecard (BSC) frameworks. Results: There was a marked increase in publications post-2019, peaking in 2023. Sub-Saharan Africa accounted for the majority of studies, with South Africa (28%) and Ghana (14%) most represented. Among the 76 included studies, the most common designs were quantitative (38%), followed by qualitative (20%), case studies (14%), and mixed-methods (11%), reflecting a predominantly empirical and field-based evidence base. Thematic analysis showed that 26% of studies focused on food safety knowledge and practices, 14% focused on infrastructure gaps, and 13% focused on policy and regulatory challenges. Of the 76 studies included, 73% reported a positive relationship between hygiene compliance and improved business performance (such as customer trust, revenue, and operational resilience), based on vote-counting across qualitatively synthesized results and business outcomes. The review identifies a conceptual synergy between the HBM’s cues to action and the BSC’s customer dimension, highlighting how hygiene compliance simultaneously influences vendor behaviour and consumer trust. Conceptual saturation was observed in themes related to hygiene protocols, consumer trust indicators, and regulatory barriers. Conclusions: Health and safety practices function not only as compliance imperatives but also as strategic assets in the informal food economy. However, widespread adoption is impeded by structural barriers including limited infrastructure, education gaps, and uneven regulatory enforcement. The findings call for context-sensitive policy interventions and public health models that align with vendor realities and support sustainable, safe, and competitive informal food systems.

## 1. Introduction

Street food vending has become an indispensable feature of urban economies, particularly in low- and middle-income countries (LMICs), where it serves as a critical source of affordable nutrition and self-employment [1]. In cities such as Accra, New Delhi, and Johannesburg, street vendors play a central role in food access for low-income populations while also contributing to urban vibrancy and public safety [2,3,4]. In South Africa, it is estimated that street food vendors contribute significantly to urban food security, with approximately 60% of urban households relying on street food for their daily meals [5].

Despite its economic and social importance, the street food sector is often characterized by unsanitary conditions, irregular enforcement of regulations, and a lack of access to infrastructure [6,7]. These deficiencies raise significant public health concerns, particularly the risk of foodborne illnesses, which are often exacerbated by poor personal hygiene, inadequate waste management, and insufficient knowledge of safe food handling [8,9,10].

To underscore the significance of the issue, it is essential to highlight specific examples and statistics. Although precise prevalence rates vary, numerous studies consistently report high levels of microbial contamination in street food across South Africa and other LMICs, posing a significant risk of foodborne illness [11,12,13]. Additionally, multiple studies highlighted poor managerial skills, recurring hygiene violations, and the presence of harmful pathogens in informal food settings, highlighting that inadequate access to water and sanitation facilities is prevalent among vendors [5,11,12,13]. These statistics underscore the urgent need for improved health and safety practices in the informal food sector.

In South Africa, vendors are legally required to comply with health protocols as per the Foodstuffs, Cosmetics, and Disinfectants Act, and the Regulation “R638” in South Africa pertains to the General Hygiene Requirements for Food Premises, Transport of Food, and Related Matters [14,15]. However, enforcement is often fragmented and punitive rather than supportive, contributing to low compliance levels [16,17,18]. Many vendors cite financial constraints, inadequate training, and a lack of access to water and sanitation as primary reasons for non-compliance [3,4,19]. While a growing body of literature acknowledges the importance of food safety in informal economies, the link between health and safety practices and business performance remains underexplored. Existing studies tend to focus either on hygiene compliance or on profitability, but rarely integrate the two using robust conceptual frameworks [5,10,11,12,13,14,15,16,17,18,19]. Furthermore, various studies have assessed street food safety, but a few have systematically synthesized how informal vendor health risks intersect with food safety governance and policy frameworks in LMICs. This review addresses this gap by focusing on multi-level determinants and linking them to implementation and monitoring frameworks.

This review further employed two complementary conceptual lenses, including the Health Belief Model (HBM) [20,21,22] and the Balanced Scorecard (BSC) framework [23]. The combined use of the HBM and BSC was intentional to provide a dual lens that captures both individual-level behavioural motivations (via the HBM) and systemic, operational, and performance perspectives (via the BSC).

This review aligns with the United Nations Sustainable Development Goals (SDGs), particularly Goal 3 on good health and well-being and Goal 8 on decent work and economic growth, which emphasize the importance of safe working environments and public health systems [24]. It is aimed at critically examining how health and safety practices influence business performance among street food vendors in both global and South African contexts. By employing the HBM and the BSC frameworks, the review seeks to provide a comprehensive understanding of the multi-level determinants affecting vendor compliance and performance.

Research Questions:(1)How do health and safety practices influence the business performance of street food vendors, particularly in South Africa and other low- and middle-income countries?(2)What are the key barriers to health and safety compliance among street food vendors, and how do these barriers vary across different contexts?(3)How do individual perceptions and motivations (as captured by the HBM) interact with systemic and operational factors (as captured by the BSC) to shape health and safety practices among street food vendors?(4)What policy interventions and support mechanisms can enhance both vendor performance and consumer safety in informal food economies?

This integrative review, therefore, offers a novel contribution by merging behavioural and operational theories to better understand compliance dynamics and performance outcomes in informal street food economies. To support this objective, the paper proceeds as follows: Section 2 outlines the integrative review methodology, including the search strategy, inclusion criteria, and conceptual frameworks. Section 3 presents the results, thematically organized, and contextualized with comparative insights from South Africa and other LMICs. Section 4 discusses these findings through the lens of the HBM and BSC, incorporating additional theoretical perspectives where relevant. Section 5 concludes by highlighting key implications, study limitations, and directions for future research and policy innovation.

## 2. Materials and Methods

### 2.1. Review Design

This study employed two methods, an integrative review methodology and the Preferred Reporting Items for Systematic Reviews and Meta-Analyses [25]. The integrative review methodology, as outlined by Whittemore and Knafl (2005), is used to systematically gather, evaluate, and synthesize diverse types of the literature, including theoretical papers, empirical studies, and policy reports related to health and safety practices in the street food vending sector [26]. The integrative approach was chosen for its ability to accommodate varied methodologies and offer a comprehensive understanding of complex, multifaceted issues such as vendor compliance and business performance. PRISMA was used for the screening of published articles. To guide thematic analysis and interpretation, two conceptual frameworks were adopted: the HBM and the BSC [20,23]. These were used to structure findings around health behaviour motivations and business performance dimensions, respectively.

### 2.2. Frameworks That Guide the Study

The dual use of the HBM and the BSC allowed the review to bridge micro-level behavioural insights with macro-level structural enablers (Figure 1). The HBM is a psychological model used to explain and predict health-related behaviours, particularly in preventive health contexts. It comprises six key constructs: (1) perceived susceptibility, (2) perceived severity, (3) perceived benefits, (4) perceived barriers, (5) cues to action, and (6) self-efficacy. These constructs help in understanding vendors’ risk perceptions, motivations, and decision-making related to food safety and hygiene practices [20]. The HBM is applied to explore individual-level perceptions of risk, benefits, and behaviour change among street vendors.

The BSC, originally developed for strategic performance management in organizations, provides a multi-dimensional lens for assessing systemic and institutional factors. It comprises four interlinked dimensions: (1) financial performance, (2) customer/stakeholder value, (3) internal processes, and (4) learning and growth (capacity building). Applied to informal food vending, the BSC enables the mapping of structural and policy-level influences that shape vendor environments, capacity, and sustainability [23]. The BSC framework was selected for its multidimensional lens, which complements the HBM by capturing operational, financial, and learning dimensions that are central to informal vendor ecosystems. Its inclusion enables the structured categorization of systemic and institutional factors affecting vendor compliance and health outcomes, elements that are often overlooked in behaviour-only models like the HBM.

These frameworks are complementary as the HBM identifies behavioural motivators and barriers, while the BSC situates them within systemic health governance and performance metrics. While the HBM captured personal motivators and deterrents, the BSC mapped the institutional terrain shaping vendor actions, including financial pressures, service delivery gaps, and capacity-building deficits. This layered analysis was especially effective in uncovering how hygiene practices are influenced not only by risk awareness but also by licensing regimes, infrastructure, and municipal engagement. This approach aligns with the multidimensional nature of informal food vending, which involves both personal risk perceptions and business-level challenges. Despite extensive research on food hygiene or vendor profitability, few studies integrate these dimensions through established frameworks. This review addresses that gap by applying both the HBM and the BSC to explore how health and safety practices affect business outcomes. The review’s unique contribution lies in bridging behavioural health theory and performance measurement in the context of informal street food vending, with a focus on both global and South African evidence.

### 2.3. Search Strategy

A systematic search was conducted between June 2024 to May 2025 across the following databases: PubMed, Google Scholar, Science Direct, Scopus, and JSTOR. The search was limited to English-language publications from 2015 to 2025, reflecting the last 10 years of evolving policy, regulatory shifts, and empirical findings in street food vending and public health domains. The Boolean search strategy was designed to capture studies at the intersection of public health and informal vendor economics. The following terms were used in various combinations across databases:Street food vendors OR informal food economy AND;Health practices OR food safety OR hygiene compliance AND;Business performance OR profitability OR revenue OR customer satisfaction AND;South Africa OR urban informal economy OR low- and middle-income countries OR LMICs.

Additional sources were identified through backward citation tracking and the grey literature from the World Health Organization (WHO), the Food and Agriculture Organization (FAO), and municipal reports from South Africa.

### 2.4. Inclusion and Exclusion Criteria

Studies were selected for inclusion based on a predefined set of criteria to ensure relevance, quality, and contextual appropriateness. Eligible studies met the following requirements: they focused on street food vendors or the broader informal food economy; reported on issues related to health, hygiene, food safety practices, or business operations; were conducted in South Africa or other LMICs; and were written in English. Furthermore, articles or grey literature sources were included if they were published between 2015 and 2025. This range was selected to ensure contemporary relevance to post-2015 global health and development frameworks (e.g., Sustainable Development Goals). These non-peer-reviewed materials were evaluated for relevance and cited to support thematic framing or comparative insight.

To maintain empirical rigour, studies were excluded if they (a) focused solely on foodborne illnesses without linking to vendor practices; (b) centred exclusively on consumer behaviours with no discussion of vendor dynamics; (c) were editorials, opinion pieces, or lacked empirical or theoretical grounding; (d) focused on high-income countries without relevance to LMIC settings; or (e) were duplicate entries retrieved across multiple databases.

Beyond a certain threshold, additional studies contributed minimal new information, signalling thematic saturation in the dataset. This point was determined not only globally but also within country-specific subsets to prevent the overrepresentation of nations with numerous similar studies. Once recurring patterns such as hygiene knowledge gaps, regulatory enforcement challenges, and infrastructural barriers emerged consistently, only the most conceptually rich and methodologically sound studies were retained. Country-specific saturation ensured that findings were not skewed by overrepresented contexts, particularly South Africa and Ghana. This approach allowed for more equitable weighting across LMICs and avoided drawing generalizations from repetitive patterns within a single setting. It further ensured a balanced yet comprehensive synthesis of the evidence base [27].

### 2.5. Study Screening and Selection

The screening and inclusion process followed the PRISMA 2020 guidelines, and a full depiction of the selection pathway is presented in Appendix A. This figure outlines the number of records identified, screened, excluded, and ultimately included in the review, ensuring the transparency and reproducibility of the process. A total of 395 records were initially identified through a combination of structured database searches and Supplementary Sources (Appendix A). Of these, 338 were retrieved from five academic databases, while an additional 57 were identified through other sources, including websites (*n* = 22), organizational repositories (*n* = 11), citation searches (*n* = 6), and manual methods such as scanning reference lists (*n* = 18). After removing duplicates, 306 records were retained for title and abstract screening. This initial screening phase was conducted independently by two reviewers, and studies that clearly failed to meet the inclusion criteria were excluded.

Following title and abstract screening, 122 full-text reports were sought for detailed evaluation. Of these, 114 were successfully retrieved and assessed for eligibility, while eight could not be accessed due to document unavailability or restrictions. After full-text assessment, 76 studies met the predefined inclusion criteria and were retained for the final synthesis. The majority of included studies (*n* = 73) originated from peer-reviewed academic journals, while a small subset (*n* = 3) were sourced from grey literature repositories, such as institutional reports or non-indexed theses.

#### Quality Appraisal of Included Studies

To support methodological rigour, each study was assessed using adapted criteria from the Joanna Briggs Institute (JBI)checklist [28]. The appraisal considered criteria such as the clarity of inclusion criteria, the appropriateness of data collection methods, rigour in data analysis, ethical reporting, and the relevance to the review objectives.

Approximately 70% of the studies were assessed to be of moderate to high methodological quality. These studies clearly defined their research aims, used valid and reliable data collection tools, and applied appropriate analytical methods. Survey-based and cross-sectional studies typically performed well on clarity of sampling procedures and measurement alignment, though some lacked detailed reporting on response rates or confounder control. Qualitative and case study research showed strength in contextual depth and participant relevance but often lacked reflexivity, triangulation, or transparency on researcher positionality. A smaller subset of studies, including theses and the grey literature, demonstrated limited methodological detail but contributed valuable contextual insights and local knowledge.

Overall, while the methodological rigour varied, the majority of studies met key standards for inclusion in the review.

### 2.6. Data Extraction and Synthesis

Thematic synthesis was conducted in two stages. First, findings were grouped according to recurring themes, such as vendor hygiene, regulatory environments, barriers to compliance, and customer perception. Then, using the HBM, studies were analyzed to assess how perceptions (susceptibility, severity, benefits, barriers, cues to action, and self-efficacy) shaped vendor behaviours [20,21,22]. In parallel, the BSC framework guided the mapping of business outcomes along financial, customer, internal process, and learning/growth dimensions [23]. The initial codebook was developed inductively through the line-by-line coding of five diverse studies, followed by iterative refinement. Two reviewers independently applied codes to each study, and discrepancies were resolved through discussion or arbitration by a third reviewer. All qualitative coding was managed using ATLAS. ti software (v.24) to support traceability and thematic saturation. Key data were extracted into a structured matrix capturing the author(s), year, and country of study; type of study (empirical, theoretical, policy-based); focus area (e.g., hygiene practices, regulatory enforcement, vendor performance); conceptual framework (if any), and key findings related to compliance and business outcomes. Manual analysis and use of Atlas Ti were used to extract word clouds [29]. Given the diverse study designs, outcomes, and reporting formats across the included literature, a meta-analysis was not feasible. Instead, a narrative synthesis was conducted using vote-counting to identify the frequency of findings linking hygiene compliance to business performance.

In addition to the extracted articles, Supplementary Literature, was incorporated where relevant to support contextual interpretation and enhance the thematic depth of the synthesis. These sources helped enrich the discussion on regulatory environments, policy gaps, and interdisciplinary frameworks, especially in areas where empirical data was limited or required triangulation [1,2,3,4,5,6,7,8,9,10,11,12,13,14,15,16,17,18,19,20,21,22,23,24,25,26,27,28,29,30,31,32,33,34,35,36,37,38,39,40,41,42,43,44,45,46,47,48,49,50,51,52,53,54,55,56,57,58,59,60,61,62,63,64,65,66,67,68,69,70,71,72,73,74,75,76,77,78,79,80,81,82,83,84,85,86,87,88,89,90,91,92,93,94,95].

## 3. Results

This section presents the synthesized findings of 76 studies included in the integrative review [2,3,4,5,6,7,9,10,11,12,13,16,17,18,19,30,31,32,33,34,35,36,38,39,40,41,42,43,44,45,46,47,48,49,50,51,52,53,54,55,56,57,58,59,60,61,62,63,64,65,66,67,68,69,70,71,72,73,74,75,76,77,78,79,80,81,82,83,84,85,86,87,88,89,90] as per Appendix A. The analysis is organized into two major segments, (1) descriptive-demographic insights and (2) thematic synthesis, reflecting both the characteristics of the reviewed studies and the conceptual insights gained. Each theme is analytically connected to the HBM and BSC frameworks [20,21,22], where applicable. These features allow us to contextualize the scope and saturation of the literature and identify underexplored areas that are relevant to street food vendor safety and performance.

### 3.1. Descriptive Characteristics of Included Studies

This subsection describes the distribution of the included studies across time, geography, study types, conceptual frameworks, and thematic areas. Quantitative frequencies and percentages are reported to provide a snapshot of the current literature trends and coverage.

#### 3.1.1. Temporal Distribution of Included Studies

This integrative review encompassed 76 studies published between 2015 and 2025 (Table 1). The temporal analysis reveals a steady growth in scholarly attention to hygiene compliance and business outcomes in informal food economies over the past decade. The highest number of studies was recorded in 2024 (18%), followed closely by 2021 and 2023 (each contributing 15%), indicating an intensified focus in the post-pandemic period. Notably, research activity began to rise significantly from 2019 onward, reflecting broader public health and policy interest in informal street food vending systems. Earlier years, such as 2015, accounted for only 1% of studies, while moderate contributions were observed in 2016–2018 (7% each), suggesting an initial but gradually expanding body of literature. All 76 studies had clearly reported publication years, ensuring consistency in trend analysis and strengthening the review’s temporal credibility.

#### 3.1.2. Geographic Distribution

Studies were drawn from a wide range of geographic contexts (Table 2). The included studies were geographically diverse, spanning over 15 countries. South Africa emerged as the most studied country with 21 studies (28%) [2,4,5,10,12,17,18,19,30,31,32,33,34,35,36,37,38,39,72,75,84,87], reflecting the country’s active informal food economy and its centrality in this research. Ghana followed with 11 studies (14%) [13,40,41,51,52,56,65,90,91,92,93], highlighting its vibrant street food culture and growing policy attention.

Other frequently studied contexts included Malaysia (4 studies), Nigeria (4), India (3), Ethiopia (3), and the USA (3) [3,22,44,50,54,57,60,66,74,76,78,80,81,82,83,86,88]. Furthermore, relevant research was identified from Jordan, Afghanistan, Uganda, Namibia, Bangladesh, Indonesia, Kenya, Vietnam, Thailand, Zimbabwe, Pakistan, and Zambia [16,43,46,55,63,70,72,79,82,85,89,95]. Among the reviewed studies, two were global in scope [6,7], and one systematic review specifically examined multiple LMICs [48].

This regional spread highlights a concentration of research in Sub-Saharan Africa, reinforcing its prominence in discussions on informal food economies and related public health challenges. However, the distribution also reveals a growing interest in Southeast Asia and global comparative reviews, suggesting a shift toward more integrated and transnational perspectives.

#### 3.1.3. Study Design and Methodology

The 76 studies included in this integrative review employed a wide range of research designs, reflecting the interdisciplinary nature of the topic (Table 3). The most prevalent category was quantitative research, comprising 37% of the studies. This group included structured surveys and cross-sectional studies that examined hygiene practices, vendor knowledge, customer perceptions, or business outcomes. These studies offered statistically grounded insights into compliance patterns and their relationships with performance metrics across various urban contexts.

Qualitative designs accounted for 13%, relying on interviews, focus groups, or ethnographic methods to explore the lived experiences of informal vendors. These studies provided depth, especially in understanding socio-cultural and gendered dimensions of hygiene and compliance. Mixed-methods research (8%) integrated qualitative and quantitative techniques, allowing for triangulated insights on both behaviours and outcomes. These were particularly valuable in evaluating the effectiveness of policy interventions or training programmes.

Case studies made up 13%, focusing on the in-depth analysis of specific settings or interventions, often situated in South African or other LMIC urban environments. These provided valuable localized context on structural challenges and municipal dynamics. Review articles, including narrative and systematic reviews, constituted 5% of the total and served to synthesize existing evidence across themes such as informal food governance or vendor health risk profiles.

Other categories included policy analyses and guideline assessments (4%), which examined national and municipal frameworks for food safety; economic or modelling studies (4%) that investigated cost–benefit or profitability aspects of hygiene; and conceptual or theoretical works (5%) that applied frameworks such as the HBM or the BSC.

Behavioural and observational studies (7%) provided real-time insights into hygiene practices through the direct observation of vendors. Finally, the grey literature and other sources (4%), such as graduate theses and WHO guidelines, added practical and policy-oriented perspectives that enriched the synthesis. This diversity enriches the integrative synthesis by offering both depth and breadth across contexts and inquiry types.

### 3.2. Conceptual Frameworks Used

Among the 76 included studies, only 24% (*n* = 18) explicitly applied a theoretical or conceptual framework to guide their analysis (Table 4). This suggests that a substantial majority of research in this area remains largely descriptive or exploratory, with limited use of structured analytical models. This gap has implications for the rigour, comparability, and policy relevance of findings across contexts.

Of the studies that employed frameworks, the most common were Public Health models (18%), including constructs like the Health Belief Model (HBM), risk perception, and behavioural health frameworks. These typically underpinned hygiene, safety, and compliance studies. Knowledge-Attitude-Practice (KAP) models were used in five studies (7%) to assess awareness and behavioural factors related to food hygiene. Policy and Regulatory Governance frameworks (5%) offered insights into compliance dynamics and institutional barriers.

Other models included consumer behaviours, Entrepreneurial Marketing, and Urban Informality, reflecting growing interdisciplinarity in the study of informal food economies. This diversity of perspectives indicates a growing interdisciplinary interest in informal food systems, integrating insights from public health, policy, urban studies, and behavioural sciences.

However, the limited uptake of conceptual frameworks highlights a methodological shortfall. It restricts the ability to synthesize findings across studies, draw causal inferences, and inform evidence-based policy. Future research would benefit from greater theoretical anchoring to improve explanatory depth, generalizability, and actionable insight for policy and practice.

### 3.3. Visual Synthesis of the Literature Focus

To complement the temporal, geographic, and thematic analyses, a word cloud was developed using the titles of all 76 included studies (Figure 2). A word cloud (Figure 2) was generated to visually synthesize recurring concepts across the reviewed studies.

Prominent terms like “food”, “vendor”, “safety”, “hygiene”, and “street” dominated the visualization. These keywords confirm a heavy emphasis on public health concerns, particularly foodborne illness prevention, environmental sanitation, and personal hygiene practices among street food vendors. The frequent appearance of terms such as “urban”, “informal”, and “risk” further reinforces the relevance of socio-economic and structural factors that shape the operations of informal food vendors in low- and middle-income countries.

Less frequent but important terms such as “training”, “consumer”, “policy”, and “performance” hint at emerging directions in the literature, specifically, the intersections of vendor education, customer behaviour, and regulatory interventions. However, phrases like “entrepreneurship”, “profit”, or “economic outcomes” were relatively rare, suggesting that while health and hygiene are well-covered, the business dimension of street food vending remains underexplored.

The word cloud also reflects conceptual saturation in certain themes, such as hygiene training, microbial risk, and consumer trust indicators. This saturation supports a growing consensus in the literature but also highlights the need for more diversified research into policy innovation, vendor empowerment, and market access areas that are currently underrepresented.

### 3.4. Thematic Mapping of Evidence

This subsection synthesizes qualitative insights from the reviewed literature into eleven major thematic categories (Table 5). These themes were identified by analyzing the stated aims and reported findings of the studies, and where applicable, were linked to theoretical frameworks such as the HBM and the BSC [20,21,22].

The most frequently addressed theme was food safety knowledge and hygienic practices, which accounted for 26% of all studies. This underscores a strong scholarly and policy focus on improving vendor handling practices, personal hygiene, and the awareness of contamination risks. The next most prominent themes were infrastructure gaps (14%) and policy reform and regulation (13%), reflecting the structural and governance-related barriers facing street food vendors, especially in urban environments with weak enforcement or ambiguous legal frameworks.

Vendor training and education were covered in 11% of the studies, highlighting the importance of formal and informal interventions in improving vendor knowledge and compliance. Themes related to hygiene compliance (9%) and consumer trust and perception (8%) showed increasing interest in measuring vendor behaviours and public response, particularly in the aftermath of public health crises such as COVID-19.

Less frequently studied but still notable were gendered health risks and the economic performance of vendors (5% each), as well as health service access, community-based interventions, and urban food security and informality (each around 3%). These emerging themes suggest growing attention to intersectional and systemic aspects of vendor well-being, including the role of social equity, community resilience, and informal economies in shaping food access and public health outcomes.

This thematic distribution reflects a shift toward more multidimensional and policy-relevant research, with increasing integration of health, economic, and governance perspectives on informal food vending.

#### 3.4.1. Urban Role of Street Food Vending

This theme provides a policy framing that is relevant to the study’s objective of exploring how urban infrastructure and governance environments affect informal vendors’ capacity to comply with hygiene standards and sustain their livelihoods within informal economies.

This subsection provides a policy framing that is relevant to the study’s objective of understanding how structural and institutional conditions shape vendors’ ability to comply with hygiene standards and maintain business performance within informal food systems.

Street food vending holds significant economic and nutritional value across rapidly urbanizing LMICs. Millions depend on this informal sector not only for affordable daily meals but also as a vital source of self-employment, particularly where access to formal labour markets remains limited [1]. In cities such as Accra, New Delhi, and Johannesburg, the informal street food sector enhances food security, activates urban public spaces, and sustains the livelihoods of thousands of low-income households [2,7,26,31,32,34,66,74,91].

Despite these benefits, street food vending often unfolds in precarious conditions. Vendors typically operate in densely populated, low-income neighbourhoods where infrastructure is poor and municipal oversight is minimal. In the South African context, several studies underscore the critical lack of access to clean water, sanitation, and adequate shelter among informal food vendors. These infrastructural deficiencies elevate health risks and significantly compromise food safety standards [4,5,12,18,36,87].

The economic significance, the contribution to urban food security, and the context-specific challenges in South Africa underscore the urgent need for supportive policy interventions. Acknowledging the sector’s role is crucial not only for urban planning but also for health and development strategies targeting marginalized populations.

#### 3.4.2. Relevance of Health and Safety Practices in Informal Food Environments

This theme directly supports the review’s aim of evaluating how hygiene practices influence vendor outcomes and public health protection, especially in resource-constrained informal settings.

Health and safety practices, including hand hygiene, food storage protocols, temperature regulation, and environmental sanitation, are foundational to preventing foodborne illnesses, particularly in informal food vending settings where risk exposure is high. The World Health Organization [8] and the Food and Agriculture Organization [1] identify these practices as core pillars of public health within food systems. Proper hygiene not only protects consumers from microbial contamination but also contributes to the long-term viability and trustworthiness of vendor businesses.

However, evidence from LMICs reveals significant inconsistencies in food safety compliance. Many informal vendors operate without formal training or awareness of standard safety protocols. Even among those with knowledge, implementation is frequently hindered by infrastructural constraints [6,10,18]. For example, in urban South Africa, vendors often lack access to clean water, proper sanitation, waste disposal systems, and designated food preparation areas, making it difficult to adhere to national or WHO-recommended hygiene practices [5,12,87].

The COVID-19 pandemic heightened public and policy attention toward hygiene and food safety standards, prompting several municipalities across the globe to revise existing protocols and issue new guidelines for street food vendors. However, the translation of these updated policies into informal vendor contexts has been limited, particularly in South Africa, where many informal food traders continue to operate outside the reach of formal health support systems [7,11,26]. This implementation gap underscores the urgent need for context-sensitive interventions, including mobile training units, decentralized hygiene infrastructure, and simplified, vendor-friendly compliance mechanisms to promote both public health and economic sustainability.

While health and safety practices are essential for disease prevention and consumer protection, their effective implementation in informal food economies is hindered by structural inequities and policy blind spots. Addressing these gaps requires a coordinated strategy involving municipal authorities, public health agencies, and vendor organizations to foster both compliance and capacity.

#### 3.4.3. Regulatory Context and Enforcement Practices

Understanding the regulatory and enforcement landscape is critical for meeting the study objective of assessing structural enablers and constraints influencing vendor compliance and performance in diverse LMIC contexts.

Street food regulation varies widely across national contexts, shaped by institutional capacities, regulatory philosophies, and local governance structures. Cross-national comparisons reveal that while some countries have developed inclusive regulatory frameworks, others remain hampered by fragmented enforcement and insufficient support mechanisms.

In countries like Ghana and Vietnam, governments have introduced progressive policies that link regulatory enforcement with capacity-building. These include simplified licencing systems, designated vending zones, and routine training workshops, which have been shown to enhance vendor compliance, food safety standards, and consumer trust [46,51]. For example, Ghana’s mobile training campaigns and structured certification processes have improved both vendor livelihoods and public confidence, offering a model of inclusive regulation that balances oversight with empowerment [51,92].

By contrast, South Africa’s regulatory framework, while technically aligned with international best practices, suffers from fragmented and inconsistent implementation. Under existing legislation, vendors are required to adhere to the FCD Act (1972), obtain a Certificate of Acceptability (R638 of 2018), and a municipal trading licence under the Businesses Act of 1991 [14,15,37]. However, municipal authorities often lack the resources, staffing, and interdepartmental coordination needed for effective enforcement [4,30,35].

This enforcement deficit frequently pushes vendors into informal or illegal operations. Field studies report that traders face steep licensing fees, complex bureaucratic processes, and a limited awareness of formal requirements, leading many to view compliance as inaccessible or punitive [4,34,38]. Rather than facilitating safe trading environments, South Africa’s current regulatory apparatus often acts as a barrier, inhibiting vendor formalization and public health adherence [10,36].

Comparatively, Ghana and Vietnam represent development-oriented regulatory approaches that integrate vendor needs and public health goals, whereas South Africa’s model tends to be enforcement-heavy without sufficient vendor support [46,51,89]. To address this, there is a need for localized, vendor-inclusive reform, including decentralization of licensing, mobile registration units, and embedded hygiene training as part of a more facilitative and developmental policy environment [5,30,51].

#### 3.4.4. Business Performance and Vendor Outcomes

Health and safety compliance in the informal food sector extends beyond public health benefits; it directly correlates with vendor profitability, brand reputation, and long-term business sustainability. Empirical research confirms that vendors who maintain strong hygiene standards not only reduce foodborne illness risks but also experience measurable improvements in customer retention, sales volume, and daily revenue [5,46,76,78].

To provide a clearer synthesis of the quantitative patterns embedded in the narrative, a visual summary was developed (Figure 3) to display the distribution of reported hygiene-related outcomes across the 76 included studies. The figure highlights that a majority of studies (*n* = 55, 73%) reported positive business or operational benefits associated with hygiene compliance, such as increased customer trust, improved sales, and better inspection outcomes. Approximately 40% of studies linked hygiene with regulatory compliance or brand perception, while a smaller subset (18%) emphasized public health impacts or reduced illness episodes. This overview clarifies the relative prominence of business, regulatory, and health-related benefits and reinforces the argument that hygiene compliance serves dual purposes in informal food systems, protecting public health and enhancing vendor livelihoods.

In Ghana, Nigeria, and South Africa, vendors who visibly demonstrate hygiene—such as through the use of gloves, clean uniforms, visible handwashing stations, or hygiene certificates, indicated greater consumer trust and a competitive advantage [6,50,88,90].

The BSC framework is instrumental in conceptualizing these outcomes. By linking internal operational practices (e.g., workflow hygiene, food handling protocols, personal hygiene) with external outcomes (e.g., customer satisfaction, vendor revenue, brand differentiation), the BSC supports a holistic understanding of how hygiene enhances business viability [23,45,82]. In this view, health compliance is reframed not as a regulatory cost but as a strategic investment in business performance.

Supporting this, several studies show that vendors who adopt structured cleaning schedules and comply with inspection protocols experience fewer health violations, improved inspection outcomes, and stronger customer engagement [44,46,78]. In contrast, those lacking visible hygiene practices face more consumer complaints, regulatory penalties, and reputational risk, reducing their long-term viability [12,38].

These findings align with the HBM constructs of perceived barriers, benefits, and cues to action. Many vendors weigh hygiene costs against tangible benefits like improved income, customer loyalty, and fewer disruptions [20,85]. The BSC’s customer-facing perspective shows that investment in sanitation and visible hygiene pays dividends in customer perception, loyalty, and even pricing power [23,76,88].

Ultimately, food safety compliance emerges as both a public health imperative and a market advantage, reinforcing the notion that hygiene is central to informal business performance, especially in competitive low- and middle-income country contexts.

#### 3.4.5. Barriers to Health and Safety Compliance

This subsection links directly to the study objective by unpacking the practical, financial, and systemic barriers that limit informal vendors’ ability to implement hygiene practices that are aligned with public health standards.

Despite the well-documented benefits of adhering to hygiene standards, many street food vendors face multifaceted barriers that hinder full compliance. These challenges, which are financial, infrastructural, educational, and cultural, are particularly acute in low-resource and informal urban contexts. Across the reviewed studies, financial barriers were the most frequently reported constraint to compliance (noted in 62% of studies), followed by infrastructure gaps such as water or sanitation (53%), educational or training limitations (39%), and cultural food practices (26%). This ranking highlights, that while compliance challenges are multifaceted, cost and basic access to hygiene infrastructure remain the most pressing obstacles in both South Africa and other LMIC contexts.

Financial constraints are among the most pervasive challenges. Vendors often operate with razor-thin margins, and expenses for gloves, disinfectants, signage, or sealed containers directly compete with essentials such as raw ingredients or daily trading fees. For many, even minimal hygiene-related costs are perceived as unaffordable [3,4,76,88].

Infrastructural barriers are equally critical (Table 6). Across Africa, Asia, and Latin America, studies have consistently shown that informal vendors lack basic infrastructure components, clean water, electricity, sanitation, and refrigeration, each of which is vital for food safety [2,5,6,10,12,14,15,26,30,31,32,36,37,38,39,51,52,56,58,65,72,75,84,87]. In South African cities like Johannesburg and Durban, traders frequently work in congested, under-serviced environments, often without access to handwashing stations or waste disposal mechanisms [4,6,10,29,35,36,71,83].

Knowledge and education gaps remain a widespread barrier. Many vendors have limited formal schooling and lack exposure to official hygiene protocols. Studies from South Africa, Ethiopia, Ghana, and Namibia reveal that even where policies exist, municipal systems often lack the capacity for effective outreach, education, or consistent inspection [26,27,30,44,52,95]. This often leads to informal knowledge systems and non-compliance, especially among newer vendors.

Cultural and behavioural norms also shape hygiene practices. In West Africa, for instance, traditional methods such as cooking in open spaces or storing food in uncovered containers can clash with modern regulatory standards. Vendors often inherit techniques from family traditions, and formal rules may feel irrelevant or even intrusive [40,41,51,52,56,58,59,65,90,92]. Without culturally sensitive training and meaningful consultation, regulatory interventions risk alienating rather than empowering vendors [43,60,85,88].

Climate variability and extreme weather events such as heatwaves, flooding, and water scarcity emerged as implicit contextual challenges in several studies, though few addressed them explicitly. Inadequate shelter, temperature-sensitive food storage, and the lack of climate-resilient infrastructure exacerbate hygiene risks. For instance, during rainy seasons, vendors often experience reduced customer footfall and face difficulty maintaining sanitary conditions due to runoff and contaminated surfaces. These environmental vulnerabilities are particularly acute in informal markets lacking a stable infrastructure, compounding existing compliance barriers.

From the HBM perspective, these challenges represent perceived barriers and low self-efficacy, both of which reduce motivation to adopt safer practices. Vendors may understand the risks but feel powerless to act due to cost, knowledge, or infrastructure gaps [20,85]. Similarly, within the BSC framework, these structural and behavioural impediments disrupt internal processes and weaken vendor learning, ultimately affecting customer satisfaction, business continuity, and financial sustainability (Table 6) [23,45,82].

Ultimately, improving compliance in the informal food sector is not a matter of stricter enforcement alone. It requires holistic, context-specific responses, ones that integrate vendor realities, remove infrastructure and cost-related barriers, and build inclusive hygiene education. Only then can compliance become a realistic, sustainable goal for informal food vendors in LMICs.

#### 3.4.6. Consumer Trust and Perception

This theme supports the study’s objective by examining how consumer trust indicators of hygiene influence vendor performance outcomes such as loyalty and revenue in informal food markets.

In informal street food economies, consumer perception functions as both a trust mechanism and a key performance driver. Without standardized certification systems or formal branding platforms, visible hygiene practices serve as informal proxies for food quality, safety, and vendor reliability. Consumers frequently assess vendors based on observable cues such as clean attire, glove or mask use, the presence of handwashing stations, and stall tidiness [5,45,50,53].

This pattern is particularly evident in South African cities like Johannesburg, where vendors with visibly hygienic setups attracted more loyal customers, while those lacking visible sanitation lost clientele, regardless of price or taste [10,15,26,36,37]. In such settings, hygiene becomes an informal branding mechanism, influencing consumer preference and repeat purchasing in the absence of formal quality assurance schemes.

The COVID-19 pandemic amplified these dynamics, heightening consumer sensitivity to hygiene. Post-pandemic, visible health and safety measures, such as mask-wearing, hand sanitizers, contactless payment, and even the public display of hygiene protocols, became expected. Vendors adopting these practices reported improvements in customer retention, satisfaction, and daily revenue, especially in crowded urban environments [1,53,78,82].

Furthermore, consumer behaviour operates as an informal market regulatory force. Vendors meeting hygiene expectations often benefit from positive word-of-mouth, repeat visits, and higher sales, while those perceived as unhygienic face immediate reputational damage and income loss [6,50,65,88]. This customer-driven accountability cycle reinforces the practical and economic importance of health compliance.

In essence, hygiene has evolved beyond a public health concern into a core branding strategy. In the informal food sector, where trust is built face-to-face and reputation spreads locally, visible sanitation measures are not just good practice; they are essential for survival and growth.

These insights align with the HBM construct of “cues to action,” where visible hygiene acts as a behavioural nudge, prompting customers to trust, engage, or avoid. From a BSC perspective, these cues directly influence the “customer perspective” dimension, where trust, loyalty, and satisfaction translate into tangible business performance metrics such as a higher sales volume, repeat patronage, and brand differentiation [20,23,45].

#### 3.4.7. Gendered Dimensions and Vulnerabilities

Addressing gendered risks contributes to the review’s broader goal of identifying equity-focused interventions that improve vendor health outcomes and compliance in informal street food systems.

Street food vending is a critical livelihood strategy for many women across LMICs, particularly in sub-Saharan Africa. Despite this, gender-specific vulnerabilities remain inadequately addressed in policy and programmatic interventions. Studies consistently reveal that women vendors disproportionately bear the burden of occupational health risks due to their dual roles as caregivers and income earners, coupled with limited access to sanitation infrastructure, health services, and economic protection mechanisms [5,12,18,30,31,65,72].

In the South African context, Hariparsad and Naidoo (2019) highlighted reproductive health risks faced by women traders in Warwick Junction, Durban, linking them to poor environmental conditions, prolonged exposure to pollutants, and inadequate access to clean water and sanitation facilities [19]. Sepadi and Nkosi (2023) found that women informal vendors in Johannesburg reported significantly higher levels of respiratory symptoms and exposure to airborne pollutants compared to their male counterparts, driven by their proximity to congested transit hubs and lack of protective interventions [12,30]. These gendered vulnerabilities are intensified by social marginalization, caregiving responsibilities, and the absence of gender-sensitive municipal planning.

Research from Ghana similarly emphasizes gendered disparities in access to hygiene infrastructure and training. Studies found that women vendors often lacked access to basic handwashing facilities and safe storage, and that such deficits directly undermined their compliance with hygiene protocols and affected their business performance [51,52,65,92]. Moreover, women’s lower educational attainment and limited mobility further reduce their ability to access certification programmes or benefit from municipal outreach efforts [13,30].

These disparities intersect directly with constructs in the HBM, particularly perceived susceptibility, modifying factors, and self-efficacy. Women vendors are often more aware of the health risks but feel powerless to mitigate them due to structural and social constraints. Within the BSC framework, these inequities fall within the learning and growth dimension, suggesting an urgent need for gender-inclusive training, targeted support schemes, and regulatory reform. Such changes must incorporate women’s lived realities, by, for instance, ensuring the proximity of sanitation facilities, flexible training schedules, and subsidies for hygiene equipment.

Addressing gendered gaps in vendor health and safety is not only a social equity concern but also a strategic imperative for improving public health and sustaining informal economies. Inclusive interventions that account for gendered experiences can enhance vendor well-being, increase compliance, and promote more equitable urban development trajectories.

#### 3.4.8. Models of Intervention and Innovation

This section contributes to the review’s aim of identifying actionable strategies that bridge health compliance and business performance through targeted, evidence-based public health interventions for informal vendors.

Although structural and behavioural barriers to hygiene compliance are well-documented, several studies demonstrate that targeted, context-responsive interventions can significantly improve both food safety outcomes and vendor livelihoods in the informal sector. These interventions tend to succeed when they blend regulatory guidance with education, infrastructure support, and economic incentives, rather than relying solely on punitive enforcement.

Across LMICs, empirical research reveals that improvements in vendor hygiene have often followed capacity-building programmes, mobile-based training initiatives, and the creation of designated vending spaces equipped with sanitation infrastructure. For instance, initiatives aimed at educating vendors on food safety protocols, sometimes conducted through participatory workshops or community outreach campaigns, have shown measurable results in improving daily hygiene practices and reducing health-code violations [46,51,56]. These improvements are frequently accompanied by increased public trust and vendor visibility in the marketplace, particularly when training is coupled with some form of certification or recognition [92].

Technological tools such as mobile learning platforms have also been used to disseminate hygiene guidance, especially in urban areas where vendors may have limited access to in-person training [70]. Likewise, city-led programmes to supply subsidized hygiene kits, establish communal handwashing stations, or restructure trading spaces have shown potential to reduce infrastructural barriers while promoting safe food handling [48,90].

These models of intervention underscore a global trend away from enforcement-heavy regimes and toward facilitative governance. Instead of viewing informal vendors as regulatory risks, many of these initiatives treat them as vital partners in public health promotion. This shift is particularly evident in programmes that reward compliance with public visibility or market benefits, rather than solely imposing penalties for non-compliance [45,51,90].

While South Africa’s formal regulatory framework includes progressive legislation like the FCD Act (No. 54 of 1972) and R638 on general hygiene, studies suggest that implementation remains fragmented and often lacks the necessary infrastructural and educational support for informal vendors to comply meaningfully [10,30,36]. Drawing on lessons from other urban contexts, proposed approaches for South African cities include mobile vendor training tied to licencing renewals, communal sanitation zones managed by vendor associations, and targeted micro-subsidies for hygiene inputs like gloves, masks, and disinfectants [1,56,70].

These types of interventions align with the HBM by enhancing vendor self-efficacy, reducing perceived barriers, and introducing cues to action. From the BSC perspective, they contribute to internal process improvements, customer satisfaction, and long-term financial sustainability, thus positioning public health compliance not only as a moral imperative but also as a sound business strategy for informal vendors.

### 3.5. Comparative Insights: South Africa vs. Other LMICs

A comparative synthesis of evidence from South Africa and other LMICs reveals both shared challenges and key contextual differences in informal food vending regulation, compliance, and consumer trust. While infrastructural and financial constraints appear universally, the nature of regulatory enforcement, vendor inclusion, and public health strategies vary markedly by country and region. Table 7 summarizes these comparative patterns, highlighting where constraints are shared and where governance responses diverge across LMIC contexts.

Vendor Demographics and Gendered Dimensions: Across most LMICs, women constitute a significant proportion of informal food vendors. However, their visibility in policymaking and access to resources remain limited. In South Africa, for example, women vendors often experience poor occupational health protections and lack access to water and sanitation infrastructure, particularly in high-density markets [12,19,28]. Similarly, in Ghana, Bangladesh, and Kenya, studies found limited access to gender-specific training or credit schemes for women [43,64,92]. While some countries have piloted inclusive vendor training and empowerment programmes, South Africa’s efforts remain nascent and fragmented [15,26,27,34].

Regulatory Enforcement: In many LMICs, fragmented and under-resourced regulatory environments limit effective street food oversight. South Africa’s municipal enforcement, though governed by national frameworks such as the FCD Act (No. 54 of 1972) and associated R638 regulations, suffers from uneven application across cities and districts, resulting in inconsistent compliance outcomes [10,11,30]. In contrast, Ghana, Vietnam, and Jordan have adopted integrated regulatory models that emphasize not only enforcement but also vendor support. These include mobile food safety outreach, simplified licensing, and certification schemes that have improved compliance and trust [45,57,77].

Compliance Barriers: Common barriers across LMICs include the lack of access to clean water, sanitation facilities, storage infrastructure, and hygiene materials. In South Africa, these infrastructural limitations combined with high compliance costs and insufficient training have been widely documented in cities such as Johannesburg and Durban [4,6,10,36,72,84]. Conversely, Ethiopia and Vietnam have demonstrated the benefits of partnerships between the government and civil society, which have facilitated infrastructure improvements and community-based hygiene education that lower compliance barriers [45,57].

Policy Environment and Institutional Framing: South Africa’s informal trading policy is predominantly enforcement-driven, with a focus on licensing, regulation, and relocation, often without developmental support. Studies point to frequent evictions, permit complications, and little investment in vendor empowerment [4,30,31]. Although policies such as the Businesses Act (1991) and R638 hygiene regulations provide a formal structure, implementation is sporadic and disconnected from vendor realities [10,11,36]. In contrast, participatory models in Ghana, Jordan, and the Philippines integrate trader perspectives through cooperations, vendor-inclusive planning, and performance-linked certification incentives [45,57].

Consumer Trust Mechanisms: Consumer behaviour also differs significantly. In South Africa, consumers rely heavily on visible cues, such as stall cleanliness, vendor grooming, and on-site handwashing as proxies for food safety in the absence of formal certifications [5,13,14,37]. Certification is rarely visible in outdoor markets, diminishing the role of formal assurance mechanisms. By comparison, countries such as Ghana and Vietnam increasingly utilize certification badges and public health inspection signage to build consumer trust and influence purchasing behaviour [16,45,57]. These certification systems not only improve compliance but also serve as marketing tools that enhance vendor competitiveness and public confidence.

In summary, while infrastructural and behavioural challenges are common across LMICs, regulatory response and support systems diverge widely. South Africa’s emphasis on regulation without adequate support contrasts with more facilitative models seen elsewhere. This highlights the urgent need for hybrid interventions that blend enforcement with vendor training, access to hygiene infrastructure, and co-designed policy mechanisms.

## 4. Discussion

This discussion is structured around four core analytic questions that reflect the aim of the review. Each question explores different dimensions of the evidence, including how health and safety practices influence vendor performance, the barriers to compliance, and the combined insights offered by the HBM and BSC frameworks. Subsections also examine policy and intervention pathways, as well as theoretical integrations, to provide a comprehensive understanding of informal food vending systems in low- and middle-income countries.

### 4.1. How Do Health and Safety Practices Influence the Business Performance of Street Food Vendors?

Street food vending functions as both an economic lifeline and a public health frontier in LMICs. Particularly in urban areas of South Africa, Ghana, and Nigeria, vendors contribute significantly to affordable food access and micro-enterprise development [1,6,16,34]. However, beyond livelihood sustenance, multiple studies demonstrate that hygiene practices directly influence business performance.

Vendors who maintain high hygiene standards, such as clean uniforms, gloves, visible handwashing stations, and sanitized equipment, report stronger customer retention, improved daily sales, and reduced health-related disruptions [5,6,16,20,46,78,82]. Hygiene becomes a substitute for formal branding, particularly in contexts like Johannesburg, where visual cleanliness determines vendor credibility and pricing power [26,45,53]. This aligns with the BSC framework, where internal hygiene routines strengthen external outcomes such as consumer trust and financial performance [20].

Empirical studies further confirm that improved hygiene results in positive inspection outcomes, lower exposure to regulatory penalties, and enhanced brand reputation [44,65,78]. These benefits represent not only public health gains but also tangible business outcomes, underscoring hygiene compliance as a strategic asset in informal food economies [63,67].

### 4.2. What Are the Key Barriers to Health and Safety Compliance Among Street Food Vendors Across Contexts?

Despite the clear benefits of food safety practices, street vendors in LMICs face numerous structural and contextual barriers to compliance. These include financial, infrastructural, educational, and cultural constraints.

Financially, vendors operate with limited margins and often prioritize raw stock or stall fees over sanitation supplies like gloves or sealed containers [3,4,76,88]. Infrastructural deficits including the lack of clean water, electricity, and sanitation are pronounced in cities like Johannesburg, Lagos, and Accra [2,5,6,10,12,26,30,36,37,51,52,56,65,84,87].

Educational barriers are widespread, especially among vendors with low formal education or limited municipal outreach [26,28,30,44,95]. Culturally, traditional food practices sometimes conflict with formal hygiene guidelines. In Ghana and parts of West Africa, for example, outdoor cooking and open storage practices remain prevalent, creating tension between lived practices and regulatory standards [38,39,40,41,51,52,56,58,59,60,65,90].

These barriers often co-exist, reinforcing each other and diminishing vendor self-efficacy. As explained by the HBM, when perceived barriers outweigh perceived benefits and when cues to action (like inspections or consumer pressure) are weak or inconsistent, vendors are unlikely to adopt safer practices, even when aware of the risks [17,18,85].

### 4.3. How Do Individual Motivations (HBM) and Operational Systems (BSC) Shape Health and Safety Compliance?

The combined use of the HBM and BSC frameworks provides a robust lens for understanding compliance behaviour among informal vendors. The HBM illuminates personal drivers and deterrents of hygiene behaviour, perceived risk, benefits, barriers, and cues to action, while the BSC links internal practices to external outcomes across learning, customer satisfaction, and financial viability [20,21,22,23].

Many vendors express an awareness of hygiene risks (perceived susceptibility) but report a lack of infrastructure, resources, or training (barriers), reducing their perceived self-efficacy [26,27]. Benefits such as customer retention and fewer health violations are often conceptual rather than experienced, reducing motivational follow-through. Meanwhile, cues to action, peer modelling, health inspections, and consumer expectations are inconsistently applied [52,60,64]. While perceived susceptibility was frequently referenced in qualitative narratives, particularly among women vendors who are aware of hygiene-related health risks, it was not consistently or explicitly measured using empirical tools. This highlights a broader gap in the operational application of HBM constructs in informal food economy research.

From a BSC perspective, poor sanitation infrastructure and inconsistent training undermine internal process improvements and limit learning and growth. On the customer side, hygiene visibility drives purchasing behaviour. Vendors who are unable to meet these expectations experience diminished customer trust and income [6,20,45,64].

Ultimately, these frameworks show that sustainable hygiene practices depend on addressing both behavioural and structural determinants, empowering vendors while reforming operational systems [91].

Beyond the BSC and HBM, additional frameworks further enrich the understanding of vendor behaviour. Systems Theory highlights the interdependence of hygiene practices with infrastructure, regulation, and institutional capacity. Human Capital Theory (HCT) frames vendor training and sanitation investments as inputs that yield productivity and business sustainability [92,93]. Total Quality Management (TQM) emphasizes continuous improvement in operations, hygiene routines, and service quality [94]. Meanwhile, the Theory of Planned Behaviour (TPB) emphasizes that hygiene adoption is driven by vendors’ intentions, shaped by attitudes, perceived norms, and control over behaviour. Together, these frameworks reinforce that improved health and business outcomes require both system-level reforms and individual-level behaviour change strategies [94].

#### Integrating the HBM’s ‘Cues to Action’ with the BSC’s Customer Dimension

This subsection advances the conceptual analysis by aligning behavioural triggers from the HBM with outcome-oriented performance metrics from the BSC framework.

The HBM and the BSC are rooted in distinct disciplinary traditions, public health and strategic management, respectively. However, in the context of informal food vending, key constructs from each framework converge in practice. One such intersection lies between the HBM’s “cues to action” and the BSC’s “customer dimension.”

Cues to action in the HBM refer to external or internal stimuli that prompt individuals to adopt health-related behaviours. These can include media messaging, observed practices, or visible environmental signals that trigger risk-reducing actions. In street food vending, examples include the use of gloves, handwashing stations, masks, and posted hygiene protocols, each of which can stimulate both self-protective behaviour and compliance among vendors.

Simultaneously, these same hygiene signals influence consumer perception. Within the BSC framework, the customer dimension emphasizes metrics such as customer satisfaction, loyalty, trust, and purchasing behaviour. When vendors adopt visible hygiene practices, customers interpret these as indicators of food safety, professionalism, and care. For instance, one study in Johannesburg reported that vendors displaying hand sanitizers and visibly maintaining cleanliness experienced repeat patronage and increased trust from their customers [5,36].

This dual effect highlights a strategic alignment: what acts as a behaviour trigger for vendors also functions as a market signal for consumers. Integrating these perspectives allows for a more holistic analysis of informal food systems, where public health imperatives and business performance are not at odds but mutually reinforcing. It also underscores the value of designing interventions, such as public signage, hygiene certifications, or mobile health units that simultaneously enhance vendor behaviour and consumer trust. Recognizing this synergy can help shape policies that treat informal vendors not just as health risks, but as partners in co-producing safe, sustainable urban food systems.

### 4.4. What Interventions and Policies Can Improve Vendor Compliance and Consumer Protection?

Successful interventions observed across countries demonstrate that supportive, vendor-centric models work better than enforcement-only strategies. In Ghana, municipal hygiene campaigns combined with certification and designated vending zones improved both public trust and vendor compliance [16]. Vietnam’s participatory workshops and Kenya’s mobile training apps addressed knowledge gaps and improved practices through community-based learning [45,90].

Cities like Accra and Mexico City implemented structural changes like shared sanitation points and subsidized hygiene kits, that enabled vendors to maintain cleanliness without incurring extra costs [1,89]. These interventions signal a global shift from punitive approaches to inclusive, developmental regulation.

In contrast, South Africa’s regulatory environment remains enforcement-heavy and fragmented, with vendors citing the inconsistent implementation of laws like the FCD Act and R638 regulations [10,11,13,35]. Vendors often face steep licensing costs, bureaucratic hurdles, and limited education or support, pushing many into non-compliance or informal operation [29,36].

Integrated interventions such as mobile licencing units, hygiene-linked permit incentives, and vendor-managed trading zones can bridge these gaps. When coupled with modular training and micro-grants for hygiene supplies, such strategies not only improve compliance but also enhance business sustainability [1,53,82].

Both the HBM and the BSC support this integrated approach. The HBM emphasizes the need for motivation-enhancing cues and barrier reduction, while the BSC demands infrastructure, learning systems, and customer-facing strategies that reinforce compliance as a viable business goal [20,45,82].

## 5. Conclusions

Health and safety compliance in the street food sector emerges from this integrative review not simply as a regulatory formality, but as a critical axis of business performance, public health protection, and urban food system resilience. Evidence synthesized from global and South African contexts demonstrates that vendors who maintain hygienic standards experience measurable benefits: increased customer satisfaction, improved market competitiveness, and reduced business disruption due to illness or enforcement actions [6,7,11,53,78,82]. However, these benefits remain out of reach for many vendors, not due to unwillingness, but because of enduring structural barriers. Financial constraints, a lack of basic infrastructure, inadequate training opportunities, and fragmented regulatory systems hinder compliance in even the most motivated vendors [10,32]. In the South African context, compliance is rarely a behavioural failure; it is a systemic one.

The findings also affirm the relevance of the HBM and BSC frameworks in explaining vendor behaviour and outcomes [20,21,22,23]. While the HBM clarifies the motivational dynamics behind hygiene practices, the BSC illustrates how internal improvements in hygiene translate into external gains, which are financial, operational, and reputational [20,21,22,23]. A dual-theoretical lens supports the case for integrated interventions that address both behavioural and structural dimensions of food safety.

By combining individual and systemic perspectives through the HBM and BSC frameworks, this review offers a novel lens for interpreting vendor behaviour and designing multi-level interventions in LMIC contexts. Future research should consider longitudinal and comparative designs that explicitly test the integrated HBM-BSC model and explore how systemic interventions rooted in HCT or RBV can be adapted for informal economies.

### 5.1. Policy and Practice Recommendations

#### 5.1.1. Future Use of the Models in Informal Vendor Management and Sustainability

The findings of this review highlight a pressing need for multi-scalar policy interventions that address both behavioural drivers and structural barriers to hygiene compliance in the informal food sector. Municipal-level governance structures often operate in silos, with fragmented responsibilities across licensing, health inspections, and vendor support. A coordinated, cross-sectoral strategy is essential to close this gap. For example, aligning health departments with economic development units could promote hygiene training as part of business support programmes, rather than as punitive enforcement mechanisms.

The dual application of the HBM and the BSC in this study offers valuable insights for public health governance and vendor development policy in informal economies. This integrated framework provides a nuanced understanding of both the motivational psychology behind hygiene behaviour (via the HBM) and the operational drivers of business performance (via the BSC).

From the HBM perspective, effective interventions must address vendors’ perceptions of the susceptibility and severity of foodborne illness, while also increasing self-efficacy and minimizing perceived barriers. Studies such as those by Ramos et al. (2021) and Liguori (2021) demonstrate that perceived risk and confidence are critical determinants of hygiene behaviour in low-resource settings [22,47]. Therefore, behavioural nudges, educational campaigns, and culturally adapted hygiene training should be developed collaboratively with vendors to ensure both relevance and uptake. Moreover, cues to action such as visible grading systems or community monitoring can reinforce safer behaviours without relying on enforcement alone.

The BSC perspective expands these insights by linking individual behaviour to organizational and performance-level outcomes. For example, improvements in hygiene (internal processes) can lead to increased customer satisfaction (customer dimension), while structured training enhances vendor skills (learning and growth). BSC implementation in health systems improves performance, sustainability, and evidence, which translates well into informal trading systems. By institutionalizing these links, municipalities can adopt performance-based incentives, such as reduced licensing fees, access to trading infrastructure, or recognition schemes for vendors who consistently meet hygiene standards.

Policy frameworks should promote cross-sectoral coordination, especially at the municipal level, where fragmented responsibilities across licensing, inspections, and economic support often hinder implementation. As noted by Sepadi and Nkosi (2023), conflicting regulations between national and local levels frequently delay or discourage formalization [30]. To address this, health and business departments should co-develop programmes that embed hygiene compliance within broader support services, aligning public health goals with economic empowerment.

Moreover, gender-sensitive approaches are essential, given that women constitute a significant proportion of the street food vendor workforce. Tailored interventions should address the specific vulnerabilities of female vendors, such as caregiving burdens, exposure to risk, and limited access to credit or infrastructure, as emphasized by Yakubu et al. (2023) and Hariparsad & Naidoo (2019) [11,19]. The BSC’s learning and growth perspective can guide efforts to build inclusive capacity-building programmes and mentorship networks.

Finally, structural investments are necessary. Infrastructure-related constraints, such as the absence of clean water, toilets, and waste disposal facilities, limit even the most motivated vendors from achieving compliance. These environmental barriers, recognized across multiple LMIC studies [45,50], highlight the need for governments to move beyond behaviour-change strategies and invest in enabling environments.

#### 5.1.2. General Recommendations

Building on the evidence reviewed, policy interventions should adopt a vendor-centric approach that balances public health goals with economic sustainability. Compliance cannot rely on punitive enforcement alone; vendors must be empowered to view health investments as good business. To support safe and sustainable informal food economies, a set of practical and scalable policy measures is proposed:Modular training programmes: Short, location-based courses that integrate food safety with business skills like inventory control, pricing, and customer engagement. Institutionalized hygiene training should be part of vendor registration, delivered in partnership with municipal health departments, NGOs, and public health educators.Hygiene-linked licencing incentives: Regulatory flexibility (e.g., reduced fees, permit extensions) for vendors who meet basic safety benchmarks.Shared infrastructure: Investment in communal water, waste, and cooking facilities in high-density vending zones to overcome infrastructural inequities.Micro-financing for hygiene supplies: Provision of gloves, disinfectants, protective gear, and handwashing stations through subsidized kits or vendor cooperations.Leverage mobile platforms for training delivery, certification tracking, and real-time inspection scheduling, tailored for informal traders.Launch public awareness campaigns that promote consumer support for visibly hygienic vendors and reinforce community-driven standards.Empower vendor associations to play a co-regulatory role through peer-monitoring, shared compliance resources, and feedback channels.

While findings are grounded in studies from a range of LMICs, contextual variability such as regulatory environments, gender norms, and economic infrastructures may limit direct transferability. Nonetheless, shared challenges like limited sanitation access, informal regulatory status, and consumer trust deficits suggest that many insights have broader applicability across comparable urban and peri-urban LMIC settings.

### 5.2. Study Limitations

This integrative review is subject to several limitations. First, while the search strategy was broad and inclusive, the reliance on English-language sources may have excluded relevant studies from non-Anglophone LMIC contexts. The grey literature and theses were included to enhance the representation of local policy and community-level data; however, they lacked consistent methodological quality. We addressed this by applying quality filters based on transparency and relevance, but acknowledge residual heterogeneity. The included literature demonstrated heterogeneity in study design, quality, and reporting standards, limiting the comparability of some findings. The thematic synthesis is interpretative in nature and may reflect subjective biases despite systematic coding procedures. Lastly, although the study applied the HBM and BSC frameworks to guide analysis, not all included studies explicitly aligned with these models, requiring inferential mapping by the reviewers. While several studies report perceived or observed improvements in business performance due to hygiene practices, none have quantified the return on investment (ROI) in monetary terms. This represents a critical evidence gap, particularly for designing incentive-based interventions.

### 5.3. Future Research Directions

The current body of the literature emphasizes health and hygiene compliance among street food vendors, yet there remains a notable gap in studies that assess the economic outcomes of these practices. While several papers acknowledge improved customer perception or vendor credibility, few provide robust data on revenue gains, profitability trends, or cost–benefit analyses associated with improved hygiene practices. Future research should carry out the following actions:Explore co-designed interventions that blend behavioural nudges (as guided by the HBM) with structural enablers (as structured in the BSC), particularly in underrepresented LMIC regions. Longitudinal designs and policy implementation studies are also needed to assess the sustained impact of health and regulatory interventions in informal food economies.Quantify the economic benefits of health compliance, including changes in sales, customer loyalty, and cost reductions from fewer illness-related work absences.Explore how branding through cleanliness affects vendor reputation and competitiveness, especially in environments where formal quality certification is lacking.Apply performance analysis tools such as the BSC or ROI-based evaluations to connect hygiene efforts to vendor business metrics.Use longitudinal and mixed-methods designs to capture changes in vendor income or sustainability over time as hygiene standards improve. Future research should also systematically integrate climate risk into analyses of informal food safety.Investigate differences in outcomes by vendor gender, geography (e.g., urban vs. rural), and type (mobile vs. stationary), as these contextual factors likely shape both health practices and economic trajectories.

By addressing these areas, future research can offer a more holistic understanding of the interplay between vendor health practices and business success, which is especially critical in LMICs where informal vending remains a cornerstone of both nutrition and employment. In conclusion, informal food vending is here to stay, and with the right support, it can become safer, stronger, and more sustainable. Cities that align vendor well-being with public health objectives can transform informal markets into vital engines of inclusive urban development.

## Figures and Tables

**Figure 1 ijerph-22-01239-f001:**
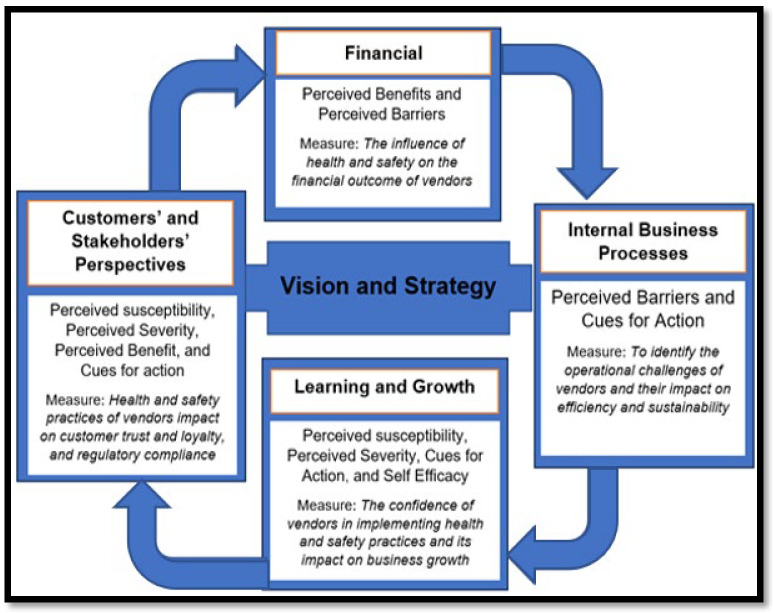
Integrated framework combining the Health Belief Model (HBM) and Balanced Scorecard (BSC) framework to assess hygiene behaviour and business performance among informal vendors.

**Figure 2 ijerph-22-01239-f002:**
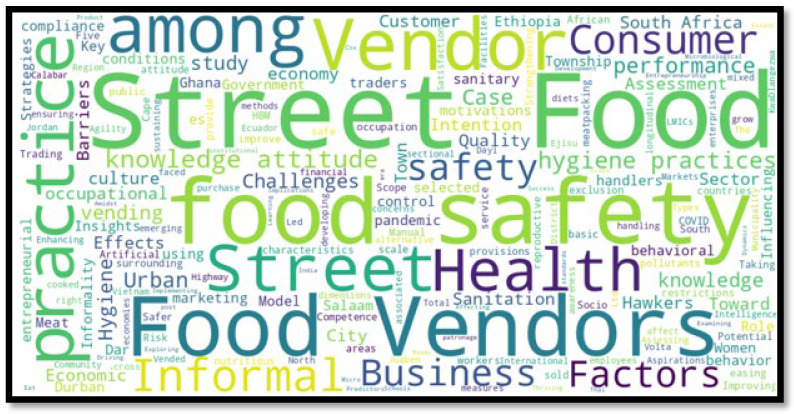
Word cloud of study titles and keywords.

**Figure 3 ijerph-22-01239-f003:**
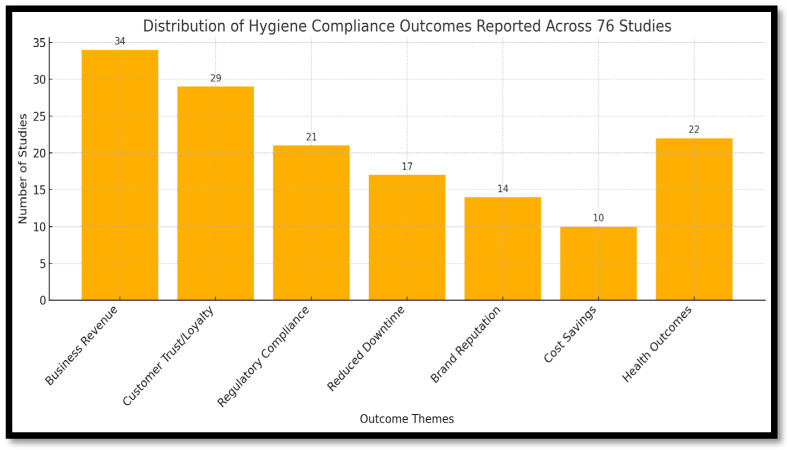
Distribution of hygiene compliance outcomes reported across 76 studies.

**Table 1 ijerph-22-01239-t001:** Number of studies per year (yearly publications).

Year	Frequency (*n*)	Percentage (%)
2015	1	1%
2016	5	7%
2017	5	7%
2018	5	7%
2019	6	8%
2020	9	12%
2021	11	15%
2022	5	7%
2023	11	15%
2024	14	18%
2025	4	15%
Total	76	100%

**Table 2 ijerph-22-01239-t002:** Publications by country.

Country	Frequency (*n*)	Percentage (%)
South Africa	21	28%
Ghana	11	14%
Nigeria	4	5%
Malaysia	4	5%
USA	3	4%
India	3	4%
Ethiopia	3	4%
Global	3	4%
Thailand	2	3%
Indonesia	2	3%
Bangladesh	2	3%
Tanzania	2	3%
Jordan	1	1%
Vietnam	1	1%
Kenya	1	1%
Uganda	1	1%
Namibia	1	1%
Zimbabwe	1	1%
Ecuador	1	1%
Oman	1	1%
Chad	1	1%
Zambia	1	1%
Afghanistan (co-listed)	1	1%
Pakistan	1	1%
Multiple LMICs	1	1%
Not Specified	2	3%

**Table 3 ijerph-22-01239-t003:** Study design breakdown.

Study Design Type	Frequency (*n*)	Percentage (%)
Quantitative (Survey/Cross-sectional)	28	37%
Qualitative (Interview/Focus Group)	10	13%
Mixed-Methods	6	8%
Case Study	10	13%
Review (Systematic/Narrative)	4	5%
Policy Analysis/Guidelines	3	4%
Economic/Modelling Study	3	4%
Conceptual/Theoretical Paper	4	5%
Behavioural/Observational	5	7%
Other (Thesis, Conference, Grey Lit)	3	4%
Total	76	100%

**Table 4 ijerph-22-01239-t004:** Conceptual framework usage frequencies.

Framework/Model	Frequency (*n*)	Percentage (%)
Public Health Models (includes HBM, Health Promotion, Behavioural Public Health, Risk Perception)	14	18%
Knowledge-Attitude-Practice (KAP)	5	7%
Policy/Regulatory Governance	4	5%
Consumer Behaviour	3	4%
Entrepreneurial Marketing	2	3%
Integrated Management Model	2	3%
Urban Informality	1	1%
Participatory Development	1	1%
Operational Constraints	1	1%
Social Equity Model	1	1%
Microeconomic Framework	1	1%
Sanitary Inspection	1	1%
Not Specified (implicit or descriptive only)	40	53%
Total	76	100%

**Table 5 ijerph-22-01239-t005:** Thematic distribution.

Theme	Frequency (*n*)	Percentage (%)
Food safety knowledge and practices	20	26%
Infrastructure gaps	11	14%
Policy reform and regulation	10	13%
Vendor training and education	8	11%
Hygiene compliance	7	9%
Consumer trust and perception	6	8%
Gendered health risks	4	5%
Economic performance of vendors	4	5%
Health service access	2	3%
Community-based interventions	2	3%
Urban food security and informality	2	3%
Total	*n* = 76	100%

**Table 6 ijerph-22-01239-t006:** Comparative summary of infrastructure-related business losses: South Africa vs. other LMICs.

Infrastructure Barrier	Reported Business Losses—South Africa	Reported Business Losses—Other LMICs
Inadequate access to clean water	Reduced food prep capacity; vendor downtime; lower customer retention	Shortened operating hours; loss of hygiene credibility
Lack of electricity	Inability to preserve perishables; spoilage-related financial losses	Food wastage; inability to store cooked food, reduced product range
Absence of sanitation facilities	Penalties during inspections; forced vendor relocation	Loss of customer trust; temporary closures due to contamination risks
Limited waste disposal services	Increased pest exposure and odour, leading to vendor fines and loss of clientele	Disease outbreaks, causing long-term customer aversion
Poorly designed vending spaces	Reduced foot traffic and revenue due to unfavourable location infrastructure	Inability to comply with distancing or hygiene norms; regulatory shutdowns

**Table 7 ijerph-22-01239-t007:** Summary of key comparative dimensions.

Dimension	South Africa	Other LMICs
Regulatory Enforcement	Fragmented and inconsistent across municipalities; often punitive in nature.	Some are punitive; however, others are more integrated and development-oriented, combining enforcement with vendor education and certification.
Vendor Demographics	Women dominate in vending, but women have been shown to experience more health impacts.	Women are often dominant in vending.
Compliance Barriers	Lack of infrastructure, financial constraints, weak municipal support, and limited training access.	Similar challenges exist, including infrastructure and finance barriers, but more access to donor-funded training programmes and government outreach initiatives.
Policy Environment	Enforcement-heavy, less developmental; limited vendor inclusion.	Development-oriented with participatory governance and mobile outreach.
Consumer Trust Mechanisms	Relies on informal proxies like visual cleanliness, personal hygiene, or word-of-mouth reputation.	Greater reliance on visible certification, inspection signage, and formal trust signals to consumers. (e.g., food safety badges).

## Data Availability

No new data were created.

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
