# Peer review of "Health and Safety Practices as Drivers of Business Performance in Informal Street Food Economies: An Integrative Review of Global and South African Evidence"

_ijerph, 2025, doi:10.3390/ijerph22081239_

Round 1
Reviewer 1 Report
Comments and Suggestions for Authors
The manuscript provides a comprehensive review exploring the relationalship between health and safety practices and business performance in informal street food economies, with a focus on low- and middle-income countries (LMICs) and South Africa. The review is timely and relevant, given the critical role of street food vending in urban food security and livelihoods. The paper needs substantial revision.
Revise the introduction to define the research gap, emphasizing how this integrative review advances prior work. A concise statement of the review’s unique contribution would strengthen the rationale.
The manuscript states that 124 sources were reduced to 49 after title/abstract screening, and then 75 studies were included after full-text review, which suggests an error. Additionally, the absence of a PRISMA flow diagram, even if simplified, limits transparency. The rationale for including studies is unclear, and the criteria for selecting “illustrative and methodologically rigorous” studies during conceptual saturation need further explanation. Clarify the discrepancy in study numbers and provide a detailed explanation of the screening and selection process, ideally with a PRISMA flow diagram as a supplementary figure. Explicitly describe how “methodological rigor” was assessed during study selection. The thematic synthesis identifies key themes (hygiene compliance, consumer trust, regulatory barriers), but the analysis lacks depth in places, particularly in connecting findings to the HBM and BSC frameworks. The discussion of barriers to compliance lists financial, infrastructural, and educational challenges but does not sufficiently explore how these align with HBM constructs. Similarly, the BSC framework is underutilized in quantifying business outcomes like revenue or customer retention. Strengthen the thematic synthesis by explicitly linking findings to HBM and BSC constructs in each subsection. For example, discuss how financial constraints (a barrier) affect vendors’ perceived self-efficacy (HBM) or internal processes.
The manuscript aims to provide a dual focus on global and South African evidence, but the South African context is not consistently integrated throughout the analysis. Restructure the thematic sections to include a dedicated subsection or comparative analysis that highlights South Africa-specific findings alongside global evidence. A table summarizing key differences (e.g., regulatory enforcement, infrastructure challenges) between South Africa and other LMICs would enhance clarity. The limitations section appears underdeveloped and does not fully address potential biases in the review process, such as publication bias, language bias (English-only studies), or the heterogeneity of study designs. The theses without clear quality assessment criteria raises concerns about the robustness of included sources. Expand the limitations section to address potential biases and justify the inclusion of theses.
Comments on the Quality of English LanguageMinor revision is needed.
Reviewer 2 Report
Comments and Suggestions for Authors
Summary of the paper
The Authors explore the relationship between health and safety practices and business performance among informal street food vendors, focusing on global and South African evidence. In the Introduction section, the Authors provide the importance of the topic, and justification of study. Also, they state the research objective and questions supported by concise literature review, and contribution of the study. In the Materials and Methods section, they detail the study methods and publication selection process, in particular, the search strategy, inclusion and exclusion criteria, and screening. Also, they detail data synthesis and extraction approach, and provide limitations and contributions of methodology employed. In the Results section, they clearly present descriptive analysis on 75 studies selected for the study. Also, they provide concise literature review on topics relating to the study. In the Discussion section, the Authors provide a discussion on three topics based on evidence from selected studies: Street Food Vending at the Intersection of Public Health and Livelihoods, How Health and Safety Practices Influence Vendor Performance, and Policy, Perception, and Conceptual Pathways to Sustainable Vendor Support. In the Conclusion section, the Authors provide a concise summary of the findings, future research topics, and policy advice.
Overall evaluation and key concerns
The paper is well-written and informative. It provides interesting reading on a key issue in the agri-food system – Health and Safety Practices and Business Performance among informal street vendors. The main contribution lies in the comprehensive methodology, allowing the Authors to conclude that Health and Safety compliance in the street food sector is not simply a regulatory formality but a driver of business performance, public health protection, and urban food system resilience. Given that street food vending and public health compliance is a subject of continuous research, and policy discussions, carrying out this analysis is apt.
However, some important issues must be addressed before the manuscript can be considered for publication. I summarize my concerns in the following points:
Introduction.
- Authors write “Existing studies tend to focus either on hygiene compliance or on profitability, but rarely integrate the two using robust conceptual frameworks [5,16].” I suggest providing more references to support your statement. In my opinion, two references are not enough, especially when your study is an integrated review.
- I suggest deleting the sub-heading “1.1. Aim and Questions of Review” and merge the text under this heading with that under the Introduction section.
- The Authors might want to provide the structure of the paper at the end of this section to give readers a fair idea of the content of subsequent sections. A couple of sentences describing the organisation of the paper would be enough.
Materials and Methods.
- Authors write in Line 78 “This study employed two methods, integrative review methodology […]” Also, in Line 79-80 they write “This study employed an integrative review methodology […]. The beginning of these two sentences appears to be repetitive. I suggest reframing one of them.
- Line 91, Authors write “A systematic search was conducted between 2024 and 2025 […]”. Between 2024 and 2025, there is no year. You may want to state the “starting month” in 2024 and the “ending month” in 2025. In this way, you can use “from” instead of “between”. This also applies to the Abstract of the manuscript.
- Line 109, change “F” in Focused to lower case.
- Authors might want to provide a flow diagram illustrating publication selection process. Although they provide detailed description of publication selection process, and in line 125-126 they provide justification for not providing a formal PRISMA diagram. A simple flow diagram could summarize the text in a concise manner.
Results.
- Authors claim that 75 studies published between 2015 and 2025 were used. However, table 1 provides contrary information. Between 2015 and 2025 means studies from 2016 to 2024. This is not what table 1 presents. Please address this. Please apply your revision in prior sections including the abstract.
- Authors might want to explain the connection of [the text under] subsections 3.4.1, 3.4.2, 3.4.3, 3.4.5 and 3.4.7 to the study objective - i.e. explore the relationship between health and safety practices and business performance among informal street food vendors, focusing on global and South African evidence - or the research question “How do health and safety practices influence the business performance of street food vendors, particularly in South Africa and other low- and middle-income countries? “ In my opinion, the text under these sections are fit for the introduction and not the results section. Please address this, otherwise merge all the text under the named subsections into the introduction section, or reframe your research objective and questions.
- Authors write […] numerous international case studies illustrate that targeted interventions can significantly enhance hygiene compliance and business outcomes in the informal food sector. Yet they do not provide references. Please address this.
Minor Comment: Proofreading required.
Comments on the Quality of English LanguageProofreading required.
Reviewer 3 Report
Comments and Suggestions for Authors
Dear authors,
Your manuscript is well written but need corrections to enhance the quality of the article. Please have a good look on the below comments. It is just to better presentation of your research:
Abstract & Introduction
- The abstract claims 73% of studies link hygiene compliance to improved business performance, but no statistical synthesis (e.g., meta-analysis) is provided to support this. Clarify if the 73% is a vote-counting synthesis (weak evidence) or based on effect sizes. If meta-analysis was not feasible, justify the narrative approach.
- The keywords maximum could be 6 words.
- Delete Line 64: 1-1 aim and scope.
- The introduction states the review covers 2015–2025, but some cited studies (e.g., WHO 2022) fall outside this range. I propose adjust the date range or justify inclusion of key pre-2015/post-2025 references.
Methods: Search Strategy & Inclusion Criteria
- The Boolean search terms lack specificity (e.g., no terms for "LMICs" or "economic outcomes").
- Grey literature inclusion is noted but not justified—risk of bias if non-peer-reviewed sources dominate. Justify grey literature use (e.g., "to capture municipal reports absent in academic databases") and assess its impact on findings.
- Rewrite Line 116-120 to clear the aim of the sentence.
- Line 140-145 at page 4 is confusing. Rewrite it.
Results: Temporal/Geographic Distribution
- Table 1 shows 15% of studies as "Not Specified" for year, weakening temporal trend analysis. Clarify it.
- Geographic bias: 27% South Africa, 15% Ghana—findings may not generalize to other LMICs. It is better to add a limitation paragraph on regional bias and call for studies in Latin America/Southeast Asia.
Study Design & Methodology
- "Empirical research" (16%) and "survey-based" (17%) are overlapping categories; unclear differentiation. Reclassify methods for enhance this section.
- No quality assessment (e.g., Joanna Briggs Institute checklist) for included studies. Add a quality appraisal table in supplementary materials.
Conceptual Frameworks
- Only 24% of studies used a framework; the rest are "not specified." This undermines theoretical rigor. You must discuss implications of framework-free studies.
- The HBM is applied to vendors, but its "perceived susceptibility" construct is not empirically measured. Propose merging HBM’s "cues to action" with BSC’s "customer dimension" in a new conceptual model.
- Add a Conceptual Framework Diagram: Merge HBM and BSC into a visual model linking hygiene practices to business outcomes.
Thematic Synthesis: Urban Role & Hygiene Practices
- Claims about COVID-19’s impact (p. 9) relies on non-peer-reviewed sources (e.g., WIEGO 2020). Replace grey literature with peer-reviewed studies on pandemic effects.
- Overemphasis on infrastructure gaps without quantifying their economic impact (e.g., "% revenue loss due to poor water access"). It must be adding a table summarizing economic losses linked to specific barriers.
- Doble check the paragraph 260-264.
Business Performance & Consumer Trust
- The BSC’s "financial dimension" is discussed anecdotally; no studies quantify ROI of hygiene investments.
- Consumer trust data (p. 12) is South Africa-centric; needs cross-country comparison. Compare trust drivers in Ghana (certification) vs. South Africa (visual cues) to highlight policy implications.
Barriers & Policy Discussion
- Barriers (financial, infrastructural) are listed but not ranked by severity or frequency.
- A revised discussion section could position it as a benchmark for LMIC street food research.
- Policy recommendations (p. 14) lack specificity (e.g., no examples of "hygiene-linked licensing incentives").
Conclusions & Future Research
- Future directions lack gender/disaggregated data (e.g., women vendors face unique barriers).
- No mention of climate change’s impact on street food safety (e.g., spoilage risks in heatwaves). Recommend studies on "climate-resilient food storage for informal vendors."
- References must be re-arrange based on the journal style. First check for academic rigor.
The manuscript is generally well-written, but some sentences are overly complex or wordy (e.g., p. 10: "Compliance cannot rely on punitive enforcement alone; vendors must be empowered to view health investments as good business").
A few grammatical errors appear, such as missing articles ("in high-density urban markets") and awkward phrasing ("Visual cues such as clean uniforms... significantly affect consumer trust" → "Visual cues... significantly influence consumer trust").
Minor redundancies occur (e.g., "holistic, context-aware interventions" could be streamlined to "context-aware interventions"). A professional proofread would enhance readability.
I propose manuscript check by an English language editor after revision by authors.
Reviewer 4 Report
Comments and Suggestions for Authors
The manuscript titled “Health and Safety Practices as Drivers of Business Performance in Informal Street Food Economies: An Integrative Review of Global and South African Evidence” addresses a highly relevant and timely topic—how health and safety practices influence the business performance of street food vendors, particularly in LMICs and South Africa. The integrative review method is well chosen, and the authors effectively apply dual frameworks (Health Belief Model and Balanced Scorecard) to structure their findings. The use of the HBM and BSC frameworks enhances both the behavioral and performance-oriented interpretation of the literature. This duality allows the study to connect hygiene practices with tangible vendor outcomes, such as improved customer trust and financial viability. Including 75 studies, covering multiple countries and various methodologies (empirical, policy-based, and conceptual), provides a comprehensive picture of the sector. The inclusion of grey literature and municipal reports adds practical insights.
While tables and thematic mapping are present, key quantitative results (e.g., percentage of studies reporting positive business outcomes from hygiene compliance) are embedded in paragraphs. A summary table synthesizing outcomes by theme would be beneficial for clarity and visual impact. For example, on page 2, it is stated that "73% of studies reported positive links between hygiene compliance and improved customer trust, revenue, or operational stability." This data could be more impactful if highlighted in a summary chart alongside other metrics.
- The claim of thematic saturation (p.3, line 113) is mentioned but not methodologically substantiated. The criteria used to determine saturation should be briefly detailed, including whether coding frequency or conceptual redundancy was the benchmark.
- The terms “hygiene compliance” and “health and safety practices” are sometimes used interchangeably. Clarifying whether these refer to formal regulatory compliance or general hygiene behavior would strengthen conceptual clarity.
- The phrase “consumer perception” might be more rigorously described as “perceived hygiene standards” or “consumer trust proxies” to align with existing behavioral economics literature.
- Although limitations are briefly acknowledged (e.g., including non-peer-reviewed sources), a more structured assessment of methodological weaknesses—such as selection bias from grey literature or geographical overrepresentation (South Africa: 27%)—would increase the transparency of the review.
I recommend this manuscript for publication pending minor revisions. The study is conceptually valuable to the literature on informal economies, food safety, and public health. Addressing the suggested refinements—particularly in clarifying data presentation, elaborating on thematic saturation procedures, and ensuring consistency in terminology—will further enhance its clarity and academic rigor.
Round 2
Reviewer 1 Report
Comments and Suggestions for Authors
Following the revisions made by the authors, the manuscript can now be considered for publication.
Reviewer 3 Report
Comments and Suggestions for Authors
Dear Authors
You tried to enhance the quality of the manuscript. I think all of my concerns are addressed and revised.
Be Successful